# Effects of Debranching Conditions and Annealing Treatment on the Formation of Starch Nanoparticles and Their Physicochemical Characteristics

**DOI:** 10.3390/foods12152890

**Published:** 2023-07-29

**Authors:** Yen-Chun Koh, Hung-Ju Liao

**Affiliations:** Department of Food Science, National Chiayi University, No. 300 Syuefu Road, Chiayi City 600355, Taiwan

**Keywords:** rice starch, enzymatic debranching, annealing treatment, starch nanoparticles, protein corona, percent recovery, pasting properties

## Abstract

Starch nanoparticles (SNPs) have unique attributes that make them suitable for specific applications. In this study, we assessed the optimum conditions for the fabrication of SNPs from the rice starches of low- (TCSG2) and medium-amylose rice lines (TK11) using pullulanase debranching combined with annealing treatment and evaluated their physicochemical and digestion properties. The highest crystalline SNP percent recoveries of 15.1 and 11.7% were obtained from TK11 and TCSG2, respectively, under the following debranching conditions: 540–630 NPUN/g, pH 5.0, 60 °C, and 12 h. The percent recovery of the crystalline SNPs by the combined modification of the debranching and the annealing treatment with an extended annealing incubation prepared from TK11 and TCSG2 was significantly increased to 25.7 and 23.8%, respectively. The modified starches from TK11 had better percent recovery of the crystalline SNPs than those from TCSG2. They exhibited a higher weight-average molecular weight (*M_w_*) and a broader/bimodal molecular weight distribution with a higher polydispersity (*PDI*) (*M_w_* = 92.76–92.69 kDa; *PDI* = 4.4) than those from TCSG2 (*M_w_* = 7.13–7.15 kDa; *PDI* = 1.7). Compared to the native counterparts, the color analyses showed that the modified starches from TK11 and TCSG2 exhibited decreased brightness (*L**)/whiteness index (*WI*) values with marked color difference values (*∆E*) ranging between 6.32 and 9.39 and 10.67 and 11.32, respectively, presumably due to the protein corona formed on the surface of SNPs which induced the browning reaction during the treatments. The pasting properties revealed that the modified starches displayed restricted swelling power with extremely low pasting viscosities, reflecting that they were highly thermally stable. The modified starches, especially those treated with an extended annealing incubation, exhibited marked decreases in the rate and extent of digestion and estimated glycemic index due to the honeycomb-like agglomerates comprising an assembly of densely packed SNPs. The results could provide helpful information for the preparation and characterization of the crystalline SNPs for potential applications such as emulsion stabilizers for Pickering emulsion and health-promoting ingredients.

## 1. Introduction

Starch is the most abundant constituent of staple foods in the human diet, serving as the primary energy source. For nutritional merits, starch can be classified into three fractions, namely, rapidly digestible starch (RDS), slowly digestible starch (SDS), and resistant starch (RS) based on digestion rate and extent [1]. Various molecular and structural features of starch induced from the interplay of several intrinsic and extrinsic factors, such as amylose content, degree of polymerization, starch granule size and surface features, crystallinity and crystalline type, food matrices, and food processing effects, significantly impact its rate and extent of digestion [1,2].

Starch nanoparticles (SNPs) are attracting significant attention due to their novel attributes beneficial to specific applications. It has been reported that starch nanoparticles (SNPs) are highly inaccessible to digestive enzymes exhibiting a two-phase digestion process due to the densely packed structure acting as a barrier to enzymatic hydrolysis exerted by the molecular structural rearrangement and the increased short-range order [3,4,5]. In addition to their nutritional merits, starch nanoparticles (SNPs) are regarded as promising engineered nanomaterials (ENM) ascribed to their novel characteristics such as nanometric size effect, high surface-to-volume ratio, biodegradability, nontoxicity and controllable release ability of bioactive compounds, which are beneficial to food and non-food applications [3,6,7,8]. SNPs can be utilized in nanofilms to improve mechanical, thermal, and barrier properties. SNPs had high oil and water absorption capacities and could be absorbed in the interfaces of the immiscible liquids to stabilize Pickering emulsion as a green alternative to the stabilizer. SNPs can be used as nanocarriers of bioactive compounds with controlled release abilities.

It has been reported that SNPs can be fabricated through top-down and bottom-up approaches. The top-down approach for the production of SNPs utilizing starch as a precursor can be categorized into conventional methods such as acid hydrolysis and non-conventional methods such as enzymatic debranching, milling, nanoprecipitation, microemulsion, sonification, ultrasound combined with non-thermal plasma, heat moisture treatment, and high hydrostatic pressure [4,6,9]. Enzymatic debranching based on a top-down approach is one of the major means to enhance the production of starch nanoparticles by increasing the self-assembling linear short chains through cleaving the α-1, 6-linkages of amylopectin. The short linear chains can re-assemble into hydrogen-bonded double-helical nanocrystals using other treatment techniques, including recrystallization, nanoprecipitation, and microemulsion [4,6,8,10,11]. Furthermore, by combining other physical means, such as hydrothermal treatments, high static pressure treatment, and ultrasonic treatment, the crystalline domains of the SNPs can be expanded, resulting in the enhanced production of crystalline SNPs [3,4,6].

To the best of our knowledge, the effects of debranching conditions and the combined annealing treatment on the formation of SNPs have yet to be systematically investigated. The objective of this study was to examine the effect of debranching conditions (enzyme concentration, pH, debranching temperature, and time) and the combined annealing treatments with different annealing temperatures and incubation times on the production of crystalline SNPs from the rice starches of two rice lines varying in amylose content. We assessed the percent recovery of the crystalline SNPs and their pasting properties using a rapid visco analyzer, surface morphology using scanning electron microscopy, color attributes using a colorimeter, and in vitro digestion properties.

## 2. Materials and Methods

### 2.1. Materials

The polished rice kernels from two rice lines, namely, Taikeng 11 (TK11) and Taichung Sen glutinous 2 (TCSG2), were used in this study. Pullulanase (E.C. 3.2.1.41, pullulanase microbial, 1824.68 NPUN/mL), pancreatic α-amylase (10 units/mg, Sigma-Aldrich A3176) and dimethyl sulfoxide (DMSO) were obtained from Sigma-Aldrich Chemical Co. (St. Louis, MO, USA). Amyloglucosidase (AMG, 3260 units/mL) and a total starch assay kit were obtained from Megazyme International Ireland Ltd. (Wicklow, Ireland). All other reagents used in the present study were of analytical grade.

### 2.2. Rice Starch Preparation

Rice starches from the two rice lines were isolated using an alkali extraction method [12]. Briefly, the polished rice kernels were washed and steeped in 0.165% NaOH in the ratio of 1:6 at ambient temperature for 24 h prior to wet-milling by a disc mill. The slurry was neutralized by 0.1 M of HCl and centrifuged to collect the filter cake, which was then oven-dried at 50 °C for 24–30 h, pulverized to pass through an 80-mesh sieve, and stored in-packed under ambient conditions for further measurements.

### 2.3. Optimization of Enzymatic Debranching for Starch Nanoparticle Production

Rice starch slurry (10%, *w*/*v*, in diluted pH 5.0 sodium acetic buffer) was prepared and pregelatinized in boiling water for 15 min prior to autoclaving for 30 min at 121 °C. To optimize the effect of debranching conditions on the formation of crystalline SNPs, the enzymatic debranching was performed at different concentrations of pullulanase (90, 180, 270, 360, 450, 540, 630, and 720 NPUN/g of starch), debranching temperatures (45, 50, 55 and 60 °C), debranching time (4, 8, 12, 16, 20 and 24 h) and pH conditions (4.5, 5.0, 5.5 and 6.0). The mixture was then autoclaved at 121 °C for 1 h for enzyme deactivation. The samples were cooled at 4 °C for 24 h before being oven-dried, grounded, and screened through an 80-mesh sieve for further measurement.

### 2.4. Combined Modification with Annealing Treatments

The combined modification with the annealing treatments was performed according to previous studies [2,13,14,15]. The samples obtained from Section 2.3 under optimum conditions were incubated at different temperatures, namely, 4, 25, 35, 45, and 55 °C for either one- or five-day annealing. Annealed samples were oven-dried at 50 °C for 24 h and grounded to pass through an 80-mesh sieve for further analysis. The highest SNP percent recovery obtained from one-day annealed samples from TCSG2 and TK11 were denoted as 1-TCSG2 and 1-TK11, while five-day annealed samples were denoted as 5-TCSG2 and 5-TK11, respectively.

### 2.5. Percent Recovery of Crystalline SNPs

The percent recovery of the crystalline SNPs was measured as the percentage of unhydrolyzed pellets based on the weight of the native starch before modification, according to a previous study [16]. Briefly, the samples (50 mg) in 25 mL sodium acetate buffer were added with 1 mL pancreatic α-amylase (3.64 mg/12 mL, 10 units/mg, Sigma-Aldrich A3176) and 60 μL amyloglucosidase (AMG, 3260 units/mL, Megazyme) and incubated at 37 °C for 16 h in a shaking water bath. Hydrolysis was terminated by adding 50 mL of 95% ethanol, and the pellet was obtained after centrifugation. The pellets were dried at 45 °C for 24 h and weighed. The percent recovery was calculated as the weight of dried pellets divided by the weight of the native starch.

### 2.6. Measurements of Amylose, Total Starch, and Crude Protein Contents

Total starch contents were determined using a Megazyme total starch assay kit (Megazyme International Ireland Ltd.) according to an approved method AACC 76–13.01 [17]. The total starch (TS), crude protein, and amylose contents were measured according to approved AACC methods 76–13.01, 46–11.02, and 61–03.01, respectively [17]. The amylose contents of the native starches from TCSG2 and TK11 were 1.71 and 20.87%, respectively.

### 2.7. Gel Permeation Chromatography

Gel permeation chromatography measurements of the samples were performed according to a previous study [3] with slight modifications. A 4 mg starch sample was mixed with 4 mL of DMSO and heated in a water bath of 80 °C for 24 h with stirring. The dissolved sample was filtered through a 2 µm filter. A GPC column KD-806M and a guard column (Showa Denko K.K., Tokyo, Japan) were used in the system equipped with a refractive index detector (Hitachi Co., Ltd., Tokyo, Japan) and a column oven-controlled at 40 °C. The eluent system was DMSO containing 0.5 mM NaNO_3_ at a 0.6 mL/min flow rate. Standard dextrans (American Polymer Standard Co., Mentor, OH, USA) with different molecular weights (MWs) were used for calibration. The weight-average MW (*M_w_*) and weight-average degree of polymerization (*DP_w_*), and polydispersity (*PDI*) were calculated according to a previous study [2].

### 2.8. Measurement of Color Attributes

The color traits of the samples were determined using a colorimeter (NE-4000, Nippon Denshoku Industries Co., Tokyo, Japan). The CIELAB color coordinates were assessed: the *L** axis denotes lightness from 100 (white) to 0 (black); the *a** axis denotes red (positive values) to green (negative values); and the *b** axis denotes yellow (positive values) to blue (negative values). The total color difference (*∆E*) between the modified and native starches was also calculated. The total color difference (*∆E*) and whiteness index (*WI*) were, respectively, determined as follows:$$∆E=\sqrt{L{*}^{2}+a{*}^{2}+b{*}^{2}}$$
$$WI=100-\sqrt{{\left(100-L*\right)}^{2}+a{*}^{2}+b{*}^{2}}$$

### 2.9. Measurement of Pasting Properties

Pasting properties of starch sample suspensions (7%, *w*/*v*) were tested using a rapid visco analyzer (RVA, Newport Scientific, Sydney, Australia) based on the procedure described by a previous study [18] with moderate modification. The starch dispersions were subjected to the heating profile as follows: the temperature holding at 50 °C for 1 min, followed by heating from 50 °C to 95 °C and holding for 5 min, and, lastly, cooling to 50 °C with a holding time of 2 min. Heating/cooling rates were set at 6 °C/min. The pasting parameters obtained were pasting temperature (PT, °C), peak viscosity (PV), trough viscosity (TV), breakdown viscosity (BV), final viscosity (FV), and setback viscosity (SV).

### 2.10. Scanning Electron Microscopy

The surface structures of samples were observed using a scanning electron microscope (SEM, Hitachi S-3500N, USA). The starch samples coated with a gold layer were mounted on circular aluminum stubs using double-sided carbon tape. Surface structures were recorded under an acceleration potential of 15 kV at a magnification of 1000× with a working distance of 14 mm to the samples.

### 2.11. In Vitro Digestibility

Determination of in vitro kinetics of starch digestion was conducted according to the method described by previous studies [2,13,14,15]. The sample (50 mg) was added with 4 mL of sodium acetate buffer and heated at 40 °C in a shaking water bath. Hydrolysis was initiated by adding tris-maleate buffer containing 2.6 U of α-amylase, and incubation was allowed at 37 °C with moderate agitation. Aliquots of 1 mL solution were taken for different periods of time, namely, 20, 30, 60, 90, 120, and 180 min, and heated to 100 °C for 5 min to inactivate α-amylase. The mixture was added with 3 mL sodium acetate buffer containing 60 µL amyloglucosidase to completely hydrolyze the digested starch into glucose and were incubated at 60 °C for 45 min. Total glucose content was determined by a GOPOD colorimetric assay. The hydrolysis kinetics of the samples were determined based on previous studies [2,13,14,15].

The starch digestogram was assumed to follow a first-order kinetic model using Microsoft Excel SOLVER based on a nonlinear optimization algorithm [19]:(1)$$C=C_{\infty }(1-e^{-kt})$$
where *C*, *C_∞_*, and *k* are starch hydrolysis (%) at time *t* and 180 min, and the first-order kinetic rate constant, respectively. The rate and extent of digestion and estimated glycemic index were determined according to a previous study [14]. The starch fractions, namely, rapidly digestible starch (*RDS*), slowly digestible starch (*SDS*), and resistant starch (*RS*), were obtained according to previous studies [14] and calculated with the following equations:(2)$$RDS(\%)=(\frac{G_{20-}G_{0}}{TS})\times 0.9\times 100$$
(3)$$SDS(\%)=(\frac{G_{120-}G_{20}}{TS})\times 0.9\times 100$$
(4)$$RS(\%)=(\frac{TS-RDS-SDS}{TS})\times 100$$
where *G*_0_, *G*_20_, and *G*_120_ are the quantities of glucose after 0, 20, and 120 min of hydrolysis, respectively.

### 2.12. Statistical Analysis

Data obtained from three independent measurements were expressed as means ± standard deviation. Analysis of variance (ANOVA) was performed, and significant differences were tested by Duncan’s multiple range test (*p* < 0.05) using SPSS version 18.0 software (Chicago, IL, USA).

## 3. Results and Discussions

### 3.1. Effect of Enzyme Concentration and Debranching Conditions on Starch Nanoparticle Formation

Figure 1A shows the effect of pullulanase concentration on the percent recovery of the crystalline SNPs from the rice lines of TCSG2 and TK11. The percent recovery of SNPs increased as the enzyme concentration increased and reached the maximum percent recovery (11.65 and 15.10%) at the pullulanase concentrations of 540 and 630 NPUN/g of starch for TCSG2 and TK11, respectively. As shown in Figure 1A, the percent recovery slightly decreased after reaching the maximum values. The optimum concentrations of debranching enzyme for the maximum percent recovery of the SNPs were different for different rice lines with different amylose content; the debranching treatment of the rice starch from TK11 (low in amylopectin) led to a higher percent recovery than those from the TCSG2 (high in amylopectin). This could be ascribed to the short linear chains from the cleavage of α-1, 6-linkage of amylopectin with chain length unsuitable for the nucleation of double helices. Instead, the long chain from amylose could be preferable for forming double helices. It has been reported that glucans with DP < 10 are unsuitable for forming double-helical nanocrystals [20]. Nevertheless, decreases in starch nanoparticle yield were observed at the higher concentrations of pullulanase, presumably due to the saturation of the enzyme with substrate.

Figure 1B–D show the effect of pH, debranching temperature, and debranching time on the percent recovery of the crystalline SNPs. As shown in Figure 1B, the maximum percent recovery of the crystalline SNPs was obtained under pH 5.0 for these two rice lines, and the pH value deviating from 5.0 decreased percent recovery. The percent recovery of the SNPs increased with increasing debranching temperature. It reached the maximum value at 60 °C, as shown in Figure 1C. These results suggested that the catalytic activity of pullulanase was pH- and temperature-dependent. The debranching times of 4–24 h were investigated in this study. The percent recovery reached the maximum value with a debranching time of 12 h, while the debranching time deviating from 12 h resulted in decreases in percent recovery, as shown in Figure 1D. Our results were comparable to those obtained in a previous study [21], suggesting that the prolonged debranching time might not be beneficial to the formation of double-helical nanocrystals. Prolonged debranching time might lead to slowing down the yield of SNPs which could be due to the hydrolysis equilibrium reached after consumption of the substrates [22,23].

### 3.2. Effect of Debranching Combined with Annealing Treatment on Starch Nanoparticles Formation

The effect of the debranching combined with the annealing treatment on the percent recovery of the crystalline SNPs was investigated in this study. Figure 2 shows the effect of different annealing temperatures (4, 25, 35, 45, and 55 °C) on SNP yield from one-day and five-day annealing incubations. As shown in Figure 2A, the percent recovery of SNPs reached the highest value at 4 °C for the one-day annealing incubation. It markedly decreased with increasing annealing temperature, presumably due to the temperature of 4 °C favorable for nucleation. Contrarily, for the 5-day annealing incubation, as shown in Figure 2B, the highest SNP percent recovery was obtained for annealing treatment at 35 °C, and the annealing temperature deviating from 35 °C resulted in significant decreases in percent recovery. The results are comparable to those obtained from previous studies [21,24]. A viscous gel-like network induced by a low annealing temperature could inhibit chain mobility, which retarded crystal nucleation [25]. It was reported that an annealing temperature close to ambient temperature was suitable for crystalline SNP formation, probably due to the temperature being more favorable for the propagation of crystals rather than nucleation [21].

### 3.3. Total Starch and Crude Protein Contents

The total starch and crude protein contents of the native starches and the modified starches are presented in Table 1. The native starches isolated from the TCSG2 and TK11 had total starch contents of 92.36% and 90.38%, respectively, suggesting that high-purity starches were isolated from the rice kernels. The SNP fabrication treatment led to a slight decrease in total starch contents in the 90.96–87.08% range, presumably due to the loss of soluble glucans during the treatments. As shown in Table 1, the modified starches undergoing one-day and five-day annealing treatments contained a significant amount of crude protein ranging between 6.06 and 6.69%, suggesting a strong binding interaction between the SNPs and protein. It has been reported that a layer of protein can cover SNPs by forming a protein corona that can alter their thermodynamic environment due to the high surface-to-volume ratio and static quenching, which could have an impact on the gastrointestinal fate of SNPs [26,27,28].

### 3.4. Weight-Average Molecular Weight, Weight-Average Degree of Polymerization, and Polydispersity Index

Figure 3 shows the molecular weight distributions of the modified starches, and Table 2 presents the weight-average molecular weight, degree of polymerization and polydispersity. The modified starches from TCSG2 exhibited a unimodal pattern of MW distribution distinct from those from TK11, which displayed a bimodal MW distribution. The modified starches from TCSG2 had an Mw value in the 7.13–7.15 kDa range with a *DP_w_* and *PDI* of 44 and 1.7, respectively. On the other hand, the modified starches from TK11 had a much higher weight-average MW (92.69–92.76 kDa) with much larger *DP_w_* and *PDI* (572 and 4.4, respectively). However, there were no significant differences in the MW distribution patterns between the one-day and two-day annealed starches, which had approximately the same *M_w_*, *DP_w_*, and *PDI* values, as shown in Table 2. This revealed that the prolonged annealing incubation did not alter the chain length and composition of the modified starches. It has been reported that DP 10–100 was suitable for forming a junction zone for forming double-helical crystallites [29].

### 3.5. Color Parameters

The results of the color parameters of the native starches and the modified starches are shown in Table 1. Lightness values (*L**) of the native starches isolated from TCSG2 and TK11 were 96.91 and 96.85, while their whiteness indices (*WI*) were 96.74 and 96.63, respectively. This suggests that the high purity of starch was obtained from the isolation. The modified starches showed lower *L** and *WI*, and marked color differences (*∆E*) compared to the native starches. Increasing annealing incubation time led to decreases in *L** value and *WI* accompanied by increases in *a** and *b** values, as shown in Table 1. The decreased lightness/whiteness of the modified starches might be due to the Maillard reaction during the treatments resulting in the dark coloration. The differences in color parameters between the modified starches were also significant. As pointed out in Section 3.3, a protein corona could form on the surface of SNPs during the fabrication treatment, intriguing the Maillard reaction and the dark coloration.

### 3.6. Scanning Electron Microscopy

Figure 4 shows the SEM images of the native starches and the modified starches. Round-polygonal rice compound granules with approximately 5–8 µm diameters were observed. No cracks and damages were observed on the surface of native granules, but the sharpness of polyhedral granules’ edges might be disrupted by alkaline during isolation. The modified starches exhibited sponge- and honeycomb-like compact structures with irregular shapes. Previous studies have noted that the coupling of the continuous sponge structure indicated the reformation of double helices in the inner part of the structure [8,30,31,32]. A previous study [3] examined topographic images of starches, which noted an assembly of SNPs. The modified starches from the high amylose content rice line showed more extensive aggregation than those from the low amylose content rice line. The size of the agglomerates was increased with increasing annealing incubation. These compact and densely packed structures have been reported to be associated with resistance to digestion [21,23,33]. It has been reported that the conversion of loosely packed structures into more compact structures was induced by hydrothermal treatment [34,35].

### 3.7. Pasting Properties

Figure 5A,B shows the pasting profiles of the native and modified starches, and Table 3 displays the corresponding pasting parameters. Pasting properties exhibit the viscosity of the starches during heating, holding, and cooling phases, reflecting the granule swelling and water-binding capacity of starches, the thermal stability of thickening power, and the gelling ability. As shown in Figure 5A and Table 3, the peak viscosity was negatively correlated to the amylose content of the native starches. The native starch isolated from TCSG2 exhibited the highest peak viscosity, probably due to its extremely high amylopectin content that contributed to extensive granule swelling and displayed the highest breakdown viscosity, presumably due to the extensive granule swelling resulting in granule fragility, which was consistent with a previous study [36]. Conversely, the native starches from TK11 displayed higher final viscosity (FV) and setback viscosity (SV) than those from TCSG2, corresponding to their higher amylose contents. As shown in Figure 5B and Table 3, the modified starches exhibited markedly low viscosities throughout the heating, holding, and cooling phases, suggesting their low swelling power and high thermal stability due to the double-helical nanoaggregates. The results were inconsistent with previous studies [13,31], which reported that the recrystallized starch possessed an undetectable pasting temperature with no significant changes in viscosity.

### 3.8. In Vitro Digestion

Starch digestograms of the native starches and the modified starch are shown in Figure 6, and the results of first-order digestion kinetics and estimated glycemic indices (*eGI*) are presented in Table 4. The *eGI* values of the native starches in cooked form from TCSG2 and TK11 were 94.89 and 87.41, respectively, regarded as high-glycemic-index (GI) foods. Moreover, the native starches had a relatively high digestion rate constant (*k*) and hydrolysis percentage at 180 min of digestion time point (*C_∞_*) due to the amorphous gelatinized starches being extremely susceptible to digestion. The results were comparable to previous studies [37,38,39]. Starch digestibility depends on several factors and their interplay, including botanical varieties, granule size, surface area and the existence of pores, cracks and channels, degree of crystallinity, and presence of protein. As compared to their native counterparts, the modified starches from these two rice lines had relatively lower starch hydrolysis percentages throughout the course of starch amylolysis with marked decreases in *C_∞_*, *k*, and *eGI*. The five-day annealing treatment led to a significantly lower digestion rate and extent than the one-day annealing treatment. The results suggested that the modification treatment induced the formation of double-helical nanocrystals, which could increase their resistance to digestion, and the extended annealing incubation time could further increase their crystalline size and perfection, which were more resistant to digestive enzymes. The rapidly digestible starch, slowly digestible starch, and resistant starch contents are shown in Table 4. The RS content of the annealing-treated starches increased significantly compared to the native counterparts and white bread. Contrarily, the RDS content showed the opposite trend. The RS content of the annealing-treated starches increased with extending the incubation time. The RS contents of the modified starches were strongly linked to the percent recovery of the crystalline SNPs, revealing the highly enzymatic resistance of the crystalline SNPs. It was pointed out in a previous study that the formation of short-range molecular order with strengthening intramolecular hydrogen bonding during digestion could be responsible for the enzymatic resistance of starch nanoparticles [3,39,40].

## 4. Conclusions

The highest percent recovery of the crystalline SNPs from two rice lines varying in amylose content using the debranching treatment was obtained under the following optimum conditions: 540–630 NPUN/g starch, pH 5.0, 60 °C, and 12 h. The debranching combined with additional annealing treatment led to the markedly increased percent recovery of SNPs. The debranching combined with five-day annealing incubation at 35 °C led to the highest increase in percent recovery. The starches from the rice line with higher amylose content resulted in a better percent recovery of the crystalline SNPs. Our results showed that the debranching with the optimum conditions in combination with the annealing treatment enhanced the crystalline SNP production to a great extent. The molecular weight distribution and molecular uniformity of the modified starches were strongly affected by the amylose content of the starch as a top-down precursor. Color examination revealed that the modified starches exhibited decreased brightness/whiteness values with a marked total color difference value due to the protein corona formed on the surface of the SNPs inducing the browning reaction during the fabrication. The pasting profiles and the SEM images revealed that the modified starches displayed honeycomb-like structures of agglomerate, comprising an assembly of densely packed SNPs. The densely packed structure contributed to their thermal stability and resistance to digestive enzymes, substantiated by their low rate and extent of digestion and estimated glycemic index, and highly resistant starch content. The results can help to elucidate the optimum fabrication conditions of the crystalline SNPs and to characterize the physicochemical properties for their potential applications, such as emulsion stabilizers in Pickering emulsion and health-promoting ingredients.

## Figures and Tables

**Figure 1 foods-12-02890-f001:**
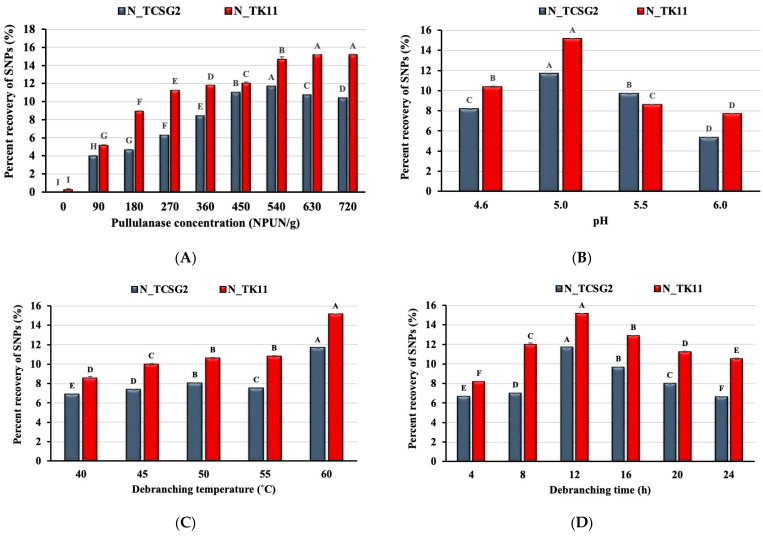
Effects of (**A**) pullulanase concentration, (**B**) pH, (**C**) debranching temperature, and (**D**) debranching time on the percent recovery of starch nanoparticles. The error bars represent the standard deviation of the mean value of three replicates. Means with different letters on each bar with different colors are significantly different (*p* < 0.05). N_TCSG2 and N_TK11 are the native starches extracted from Taichung Sen glutinous 2 (TCSG2) and Taikeng 11 (TK11), respectively.

**Figure 2 foods-12-02890-f002:**
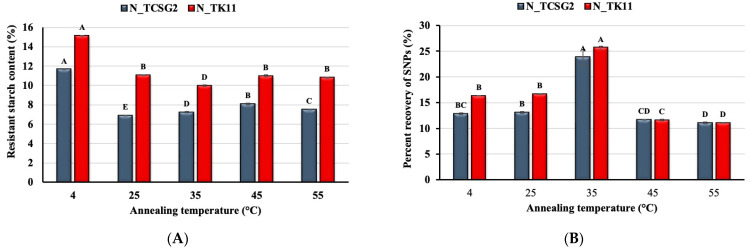
Effect of annealing temperature on the percent recovery of starch nanoparticles for (**A**) one day and (**B**) five days of incubation time. The error bars represent the standard deviation of the mean value of three replicates. Means with different letters on each bar with different colors are significantly different (*p* < 0.05).

**Figure 3 foods-12-02890-f003:**
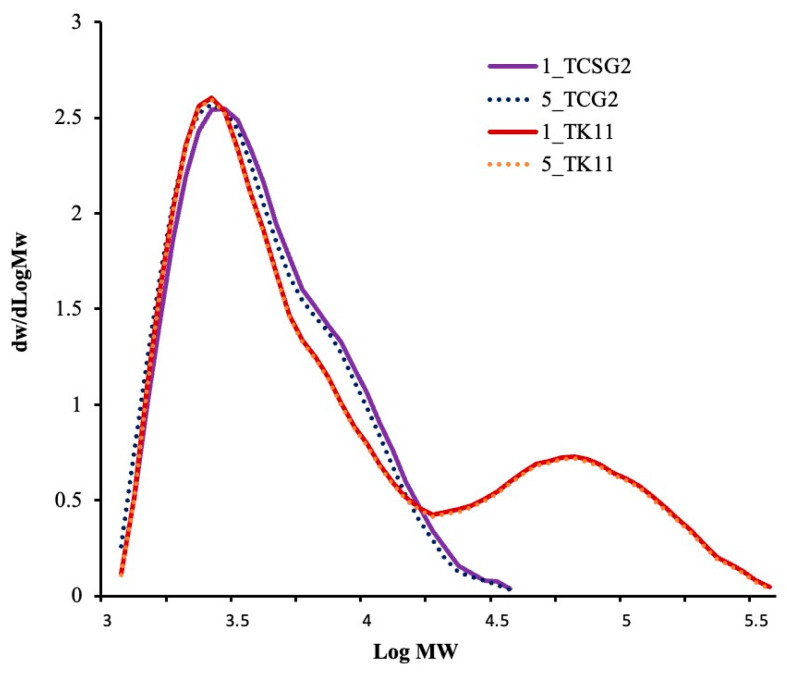
Molecular weight distributions of the modified starches.

**Figure 4 foods-12-02890-f004:**
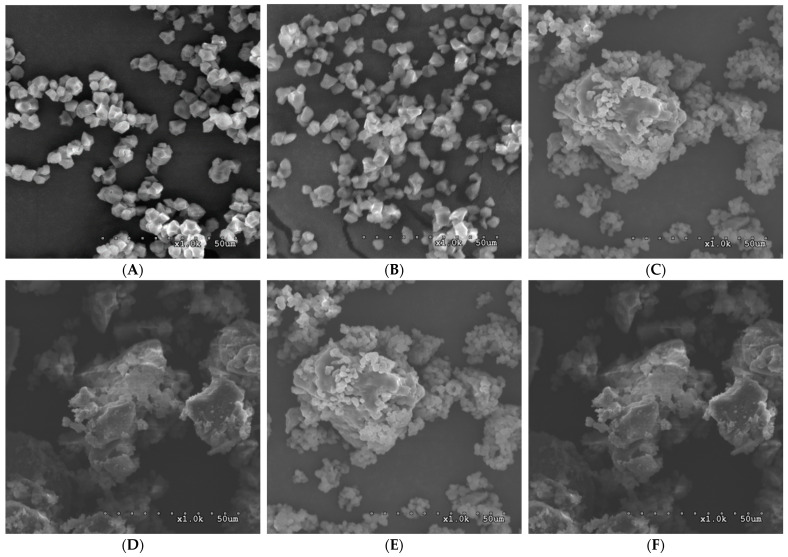
Images of scanning electron microscopy of the native starches and the modified starches, namely, (**A**) N_TCSG2, (**B**) N_TK11, (**C**) 1_TCSG2, **(D**) 1_TK11, (**E**) 5_TCSG2, and (**F**) 5_TK11 (Magnification ×1000).

**Figure 5 foods-12-02890-f005:**
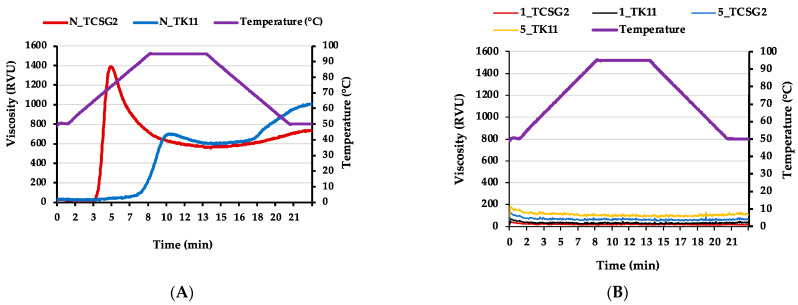
Pasting properties of the (**A**) native starches and (**B**) modified starches.

**Figure 6 foods-12-02890-f006:**
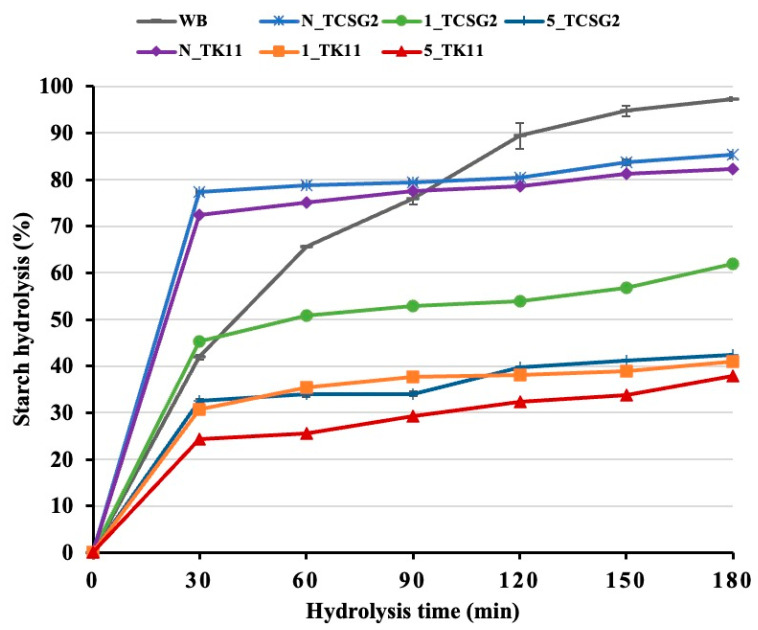
Starch digestograms of the native starches and the modified starches.

**Table 1 foods-12-02890-t001:** Total starch, crude protein contents, and color parameters of the native and modified starches.

	*TS* (%)	*CP* (%)	*L**	*a**	*b**	*∆E*	*WI*
N-TCSG2	92.36 ± 1.06 ^a^	6.22 ± 0.04 ^c^	96.91 ± 0.01 ^a^	-0.42 ± 0.006 ^e^	0.95 ± 0.006 ^f^	N.D.	96.74 ± 0.01 ^a^
1-TCSG2	90.21 ± 0.62 ^a^	6.69 ± 0.04 ^a^	92.03 ± 0.02 ^c^	0.020 ± 0.012 ^d^	8.41 ± 0.015 ^d^	6.32 ± 0.015 ^d^	88.42 ± 0.00 ^c^
5-TCSG2	90.28 ± 1.13 ^a b^	6.61 ± 0.12 ^b^	89.67 ± 0.01 ^d^	0.210 ± 0.012 ^c^	10.45 ± 0.01 ^c^	9.39 ± 0.013 ^c^	85.30 ± 0.00 ^d^
N-TK11	90.38 ± 1.13 ^a^	6.14 ± 0.04 ^c^	96.85 ± 0.02 ^b^	-0.39 ± 0.025 ^e^	1.11 ± 0.012 ^e^	N.D.	96.63 ± 0.01 ^b^
1-TK11	90.96 ± 2.32 ^ab^	6.45 ± 0.27 ^b^	88.34 ± 0.01 ^f^	0.58 ± 0.012 ^a^	11.98 ± 0.006 ^a^	11.32 ± 0.014 ^a^	83.28 ± 0.00 ^f^
5-TK11	87.41 ± 0.97 ^b^	6.65 ± 0.16 ^ab^	88.42 ± 0.02 ^e^	0.37 ± 0.020 ^b^	11.22 ± 0.030 ^b^	10.67 ± 0.003 ^b^	83.87 ± 0.02 ^e^

*TS*: total starch content. *CP*: crude protein content. *L**, *a** and *b**: CIE*L***a***b** color coordinates. *∆E*: the total color difference between the modified and native starches. *WI*: whiteness index. Data are presented as mean ± standard deviation of triplicate measurements. Means with different superscript letters within each column are significantly different (*p* < 0.05).

**Table 2 foods-12-02890-t002:** Weight-average molecular weight, weight-average degree of polymerization, and polydispersity index of the native and modified starches.

Sample	*M_w_* (kDa)	*DP_w_*	*PDI*
N-TCSG2	N.D.	N.D.	N.D.
1-TCSG2	7.15 ± 0.08 ^b^	44 ± 0 ^a^	1.7
5-TCSG2	7.13 ± 0.06 ^b^	44 ± 0 ^a^	1.7
N-TK11	N.D.	N.D.	N.D.
1-TK11	92.76 ± 0.18 ^a^	572 ± 1 ^a^	4.4
5-TK11	92.69 ± 0.74 ^a^	572 ± 5 ^a^	4.4

N.D.: not detected. Means with different superscript letters within each column are significantly different (*p* < 0.05). *M_w_*: weight-average molecular weight. *DP_w_*: weight-average degree of polymerization. *PDI*: polydispersity.

**Table 3 foods-12-02890-t003:** Pasting properties of the native and modified starches.

Sample	PT (°C)	PV (RVU)	TV (RVU)	BV (RVU)	FV (RVU)	SV (RVU)
N-TCSG2	66.2 ± 0.2 ^a^	1389 ± 23 ^a^	585 ± 9 ^b^	804 ± 7 ^a^	737 ± 5 ^b^	152 ± 2 ^b^
1-TCSG2	N.D.	N.D.	N.D.	N.D.	15 ± 1	N.D.
5-TCSG2	N.D.	N.D.	N.D.	N.D.	27 ± 2	N.D.

N-TK11	66.0 ± 0.1 ^a^	696 ± 2 ^b^	601 ± 5 ^a^	95 ± 1 ^b^	1006 ± 30 ^a^	405 ± 2 ^a^
1-TK11	N.D.	N.D.	N.D.	N.D.	22 ± 1	N.D.
5-TK11	N.D.	N.D.	N.D.	N.D.	40 ± 2	N.D.

N.D.: not detected. PT: pasting temperature. PV: peak viscosity. TV: trough viscosity. BV: breakdown viscosity. FV: final viscosity. SV: setback viscosity. Data are presented as mean ± standard deviation of triplicate measurements. Means with different superscript letters within each column are significantly different (*p* < 0.05).

**Table 4 foods-12-02890-t004:** Equilibrium concentration, kinetic constant, hydrolysis index, estimated glycemic index and starch fractions of the native and modified starches.

	*C_∞_* (%)	*K* (min^−1^)	*HI*	*eGI*	*RDS* (%)	*SDS* (%)	*RS* (%)
N-TCSG2	85.32 ± 0.58 ^b^	0.03 ± 0.00 ^a^	94.17 ± 1.24 ^a^	91.41 ± 0.68 ^a^	75.87 ± 1.30 ^a^	4.48 ± 0.04	19.65 ± 0.54 ^e^
1-TCSG2	62.02 ± 0.60 ^d^	0.02 ± 0.00 ^b^	60.85 ± 2.19 ^c^	73.11 ± 1.20 ^b^	65.80 ± 0.54 ^c^	8.97 ± 0.05 ^b^	25.21 ± 0.14 ^d^
5-TCSG2	42.41 ± 0.30 ^e^	0.02 ± 0.00 ^b^	44.43 ± 3.41 ^e^	64.10 ± 1.87 ^c^	58.57 ± 0.89 ^d^	8.07 ± 0.01 ^b^	33.34 ± 0.22 ^c^
N-TK11	82.21 ± 0.00 ^c^	0.03 ± 0.00 ^a^	92.22 ± 2.30 ^b^	90.34 ± 1.26 ^a^	70.86 ± 1.26 ^b^	7.77 ± 0.12c	21.37 ± 0.45
1-TK11	40.92 ± 0.10 ^f^	0.02 ± 0.00 ^b^	43.19 ± 3.00 ^d^	63.42 ± 1.65 ^c^	54.77 ± 1.34 ^e^	7.87 ± 0.04 ^c^	37.36 ± 0.89 ^b^
5-TK11	37.85 ± 0.21 ^g^	0.02 ± 0.00 ^b^	35.04 ± 4.01 ^f^	58.94 ± 0.86 ^d^	53.5 ± 1.24 ^f^	6.07 ± 0.09 ^d^	40.43 ± 0.67 ^a^
WB	97.29 ± 0.21 ^a^	0.028 ± 0.001 ^a^	100	100	71.17 ± 1.54 ^b^	8.31 ± 0.07 ^a^	20.50 ± 1.25 ^f^

Means (triplicate measurements) with different superscript letters within each column are significantly different (*p* < 0.05). *C_∞_*: starch hydrolysis at 180 min of digestion time point. *k*: rate constant. *HI*: hydrolysis index. *eGI*: estimated glycemic index. WB: white bread. *RDS*: rapidly digestible starch. *SDS*: slowly digestible starch. *RS*: resistant starch.

## Data Availability

Data are available on request from the corresponding author, H-J. Liao.

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
