# Peer review of "Effects of Debranching Conditions and Annealing Treatment on the Formation of Starch Nanoparticles and Their Physicochemical Characteristics"

_foods, 2023, doi:10.3390/foods12152890_

Round 1
Reviewer 1 Report
The manuscript was check and came to conclusion that what was the novelty in this work? did not found any breakthrough in this manuscript. I have thoroughly assessed the submitted work and found that it lacks sufficient novelty to meet the standards of this journal. While the topic of starch and its fractions is of importance, it is essential for published papers to contribute new and original insights to the scientific community.
some comments are here.
First of all, the paper is not as per journal template. Try to condense the information and eliminate unnecessary repetition. For example, instead of saying "Starch nanoparticles (SNPs) are attracting significant attention due to their novel attributes beneficial to specific applications," you could say "Starch nanoparticles (SNPs) have unique attributes that make them suitable for specific applications." Provide more specific details about the findings, such as the actual values obtained for the percent recovery of SNPs and the changes in color parameters. This will make the results more concrete and easier to understand. State how the results and findings of the study could be useful or impactful. For example, you could mention potential applications or industries that could benefit from the preparation and attributes of the crystalline SNPs.
Authors can modify or update the information provided about starch, its fractions, and their impact on digestion. They can also include additional relevant information or research findings.
Authors can update the references cited in the text to include the most recent and relevant studies or sources.
Authors can make changes to the language and writing style to improve clarity, coherence, and readability of the text.
Authors can reorganize or rephrase sentences and paragraphs to improve the flow and structure of the information.
The weak point is the selection of methods and materials, I think the authors did not take the publication parameters seriously. The invitro digestibility is not standard and no RS, DS, IDS which are different starches after digestion has been counted. As a whole the paper is missing novelty and such work already available in literature.
Author Response
Reviewer #1:
General comments:
The manuscript was check and came to conclusion that what was the novelty in this work? did not found any breakthrough in this manuscript. I have thoroughly assessed the submitted work and found that it lacks sufficient novelty to meet the standards of this journal. While the topic of starch and its fractions is of importance, it is essential for published papers to contribute new and original insights to the scientific community.
some comments are here.
Response:
We thank the reviewer for the time and effort in assessing our manuscript and the valuable comments and constructive suggestions which will help us improve the quality of our manuscript. The manuscript has been carefully revised (highlighted in red) to address the reviewers’ concerns fully. We have enhanced the abstract, materials and methods, results and discussion, and conclusion sections to include a more detailed discussion of the results. The starch fractions have been added to the manuscript (please see the contents in Sections 2.11 and 3.8, highlighted in red). We believe that the quality of our manuscript has been improved significantly.
Comment 1.
First of all, the paper is not as per journal template. Try to condense the information and eliminate unnecessary repetition. For example, instead of saying "Starch nanoparticles (SNPs) are attracting significant attention due to their novel attributes beneficial to specific applications," you could say "Starch nanoparticles (SNPs) have unique attributes that make them suitable for specific applications." Provide more specific details about the findings, such as the actual values obtained for the percent recovery of SNPs and the changes in color parameters. This will make the results more concrete and easier to understand. State how the results and findings of the study could be useful or impactful. For example, you could mention potential applications or industries that could benefit from the preparation and attributes of the crystalline SNPs.
Response:
We thank the reviewer for the comments. We have used the “Foods Microsoft Word template” to revise our manuscript. We have added the data of the percent recovery and color parameters to the abstract section accordingly and revised the abstract section carefully with more data to address the reviewer’s comments fully (please see the content highlighted in red in the abstract section).
Comment 2.
Authors can modify or update the information provided about starch, its fractions, and their impact on digestion. They can also include additional relevant information or research findings.
Response:
We thank the reviewer for the comments. We have added the contents regarding starch fractions to Section 2.11 and the results and discussion with references about their impacts on digestion to Section 3.8 according to the reviewer’s comments.
Comment 3.
Authors can update the references cited in the text to include the most recent and relevant studies or sources.
Response:
We thank the reviewer for the valuable comments. Additional references published after 2020 relevant to this study have been added to the manuscript according to the reviewer’s comments (please see the contents in the reference section highlighted in red).
Comment 4.
Authors can make changes to the language and writing style to improve clarity, coherence, and readability of the text.
Response:
We thank the reviewer for the comments. We have carefully checked the manuscript and used the Grammarly plug-in to improve the grammar and readability. Hopefully, it can meet the journal’s standards.
Comment 5.
Authors can reorganize or rephrase sentences and paragraphs to improve the flow and structure of the information.
Response:
We thank the reviewer for the comments. The manuscript has been thoroughly revised and edited by a native English speaker to improve the flow and structure of the information. Hopefully, it can meet the journal’s standards.
Comment 6.
The weak point is the selection of methods and materials, I think the authors did not take the publication parameters seriously. The invitro digestibility is not standard and no RS, DS, IDS which are different starches after digestion has been counted. As a whole the paper is missing novelty and such work already available in literature.
Response:
We thank the reviewer for the comments. We realized the reviewer’s concerns. We have enhanced the contents in the abstract, materials and methods, results and discussion, and conclusion sections according to the reviewer’s comments. We have included several up-to-date references relevant to our study and a more detailed discussion and presentation of our data accordingly. The manuscript has been carefully revised (highlighted in red) to address the reviewers’ concerns fully. The results regarding starch fractions, namely rapidly digestible starch, slowly digestible starch, and resistant starch, have been added. We thank the reviewer for the valuable comments and are confident they should strongly strengthen our manuscript. Hopefully, the reviewer will find our explanation justified.
Reviewer 2 Report
Detailed recommendation:
Abstract: please add more data to the abstract.
Key words: add: pasting properties
Please calculated ∆E in color analysis
Research results should be described in more detail and discussed with the literature.
In conclusion, please indicate whether this solution was favorable or unfavorable?
Author Response
Reviewer #2:
Response:
We thank the reviewer for the time and effort in assessing our manuscript and the valuable comments and constructive suggestions which will help us improve the quality of our manuscript. We have carefully revised the manuscript (highlighted in red) to address the reviewers’ concerns fully. We believe that the quality of our manuscript has been improved significantly.
Comment 1.
Abstract: please add more data to the abstract.
Response:
We thank the reviewer for the comment. We have added more data to the abstract section according to the reviewer’s comment (please see the contents highlighted in red in the abstract section).
Comment 2.
Key words: add: pasting properties
Response:
We thank the reviewer for the comment. Pasting properties have been added to Keywords.
Comment 3.
Please calculated ∆E in color analysis
Response:
We thank the reviewer for the comment. We have added ∆E in the color analysis accordingly (please see the contents in Sections 2.8 and 3.5).
Comment 4.
Research results should be described in more detail and discussed with the literature.
Response:
We thank the reviewer for the comment. We have revised the results and discussions section with more detailed discussions, and up-to-date references relevant to our study added to the manuscript.
Comment 5.
In conclusion, please indicate whether this solution was favorable or unfavorable?
Response:
We thank the reviewer for the comment. The conclusion section has been revised according to the reviewer’s comments.
Reviewer 3 Report
The article entitled "Effects of debranching conditions and annealing treatment on the formation of starch nanoparticles and their physicochemical characteristics" is well-structured, with relevant analysis and innovative techniques employed. The findings are interesting, revealing the effects of debranching conditions and the combined annealing treatment on the formation of starch nanoparticles from the rice starches that are attracting significant attention due to their novel attributes beneficial to specific applications. The work provides novel information on the physicochemical and digestion properties of starch nanoparticles, and the results can help to elucidate the optimum fabrication conditions of the crystalline starch nanoparticles and their properties for potential applications.
The abstract is written correctly.
The introduction describes the problem well. Well-chosen literature is cited.
The material and methods are described sufficiently.
The test results are correctly described.
The tables and figures are easy to read.
Discussion of results are correctly written.
The Conclusion section was supported by the results, was well written and provides a good conclusion for the study.
The literature is properly selected.
Minor comments:
Line 92: please delete the point after use
Line 223: 2.3. Effect of combined annealing treatment ….. please replace with
Effect of combined debranching and annealing treatment ….
Please verify the figures number at the end of the work; it is different from that mentioned in the text (e.g., Fig. 4 need to be Fig. 5, ....)
Author Response
Reviewer #3:
General comment:
The article entitled "Effects of debranching conditions and annealing treatment on the formation of starch nanoparticles and their physicochemical characteristics" is well-structured, with relevant analysis and innovative techniques employed. The findings are interesting, revealing the effects of debranching conditions and the combined annealing treatment on the formation of starch nanoparticles from the rice starches that are attracting significant attention due to their novel attributes beneficial to specific applications. The work provides novel information on the physicochemical and digestion properties of starch nanoparticles, and the results can help to elucidate the optimum fabrication conditions of the crystalline starch nanoparticles and their properties for potential applications. The abstract is written correctly. The introduction describes the problem well. Well-chosen literature is cited. The material and methods are described sufficiently. The test results are correctly described. The tables and figures are easy to read. Discussion of results are correctly written. The Conclusion section was supported by the results, was well written and provides a good conclusion for the study. The literature is properly selected.
Response:
We thank the reviewer for the time and effort in assessing our manuscript and the valuable comments and constructive suggestions which will help us improve the quality of our manuscript. We have carefully revised the manuscript (highlighted in red) to address the reviewers’ concerns fully. We believe that the quality of our manuscript has been improved significantly.
Minor comments:
Comment 1.
Line 92: please delete the point after use
Response:
We thank the reviewer for pointing this out. This has been revised accordingly.
Comment 2.
Line 223: 2.3. Effect of combined annealing treatment ….. please replace with Effect of combined debranching and annealing treatment ….
Response:
We thank the reviewer for pointing this out. This has been revised accordingly (please see Section 3.2).
Comment 3.
Please verify the figures number at the end of the work; it is different from that mentioned in the text (e.g., Fig. 4 need to be Fig. 5, ....)
Response:
We thank the reviewer for pointing this out. The figure numbers have been corrected accordingly.
Round 2
Reviewer 1 Report
Authors endorsed all of the suggested comments and make the manuscript fine enough. Accept in current format